# Clinical Outcomes in Adenoid Cystic Carcinoma of the Nasal Cavity and Paranasal Sinus: A Comparative Analysis of Treatment Modalities

**DOI:** 10.3390/cancers16061235

**Published:** 2024-03-21

**Authors:** Tae Hoon Lee, Kangpyo Kim, Dongryul Oh, Kyungmi Yang, Han-Sin Jeong, Man Ki Chung, Yong Chan Ahn

**Affiliations:** 1Department of Radiation Oncology, Samsung Medical Center, Sungkyunkwan University School of Medicine, Seoul 06351, Republic of Korea; taehoon42.lee@samsung.com (T.H.L.); dongryul.oh@samsung.com (D.O.); kyungmi.yang@samsung.com (K.Y.); 2Department of Radiation Oncology, Jeju National University Hospital, Jeju National University College of Medicine, Jeju 63241, Republic of Korea; kangpyo.kim@jejunuh.co.kr; 3Department of Otorhinolaryngology, Samsung Medical Center, Sungkyunkwan University School of Medicine, Seoul 06351, Republic of Korea; hansin.jeong@samsung.com (H.-S.J.); chungmk@skku.edu (M.K.C.)

**Keywords:** adenoid cystic carcinoma, nasal cavity, paranasal sinus, definitive radiation therapy, adjuvant radiation therapy

## Abstract

**Simple Summary:**

Adenoid cystic carcinoma (ACC) is known to have distinctive clinical features. It tends to metastasize to the lungs and spread through the nerves. As the nasal cavity and paranasal sinuses are close to cranial nerves, ACC arising in such areas presents explicit challenges for treatment. This study reported the treatment patterns and outcomes of sixty-one patients with sinonasal ACC who were treated at a single institution. Patients with more extensive disease underwent radiation therapy as the mainstay of treatment, and these showed worse treatment outcomes when compared to those who were able to undergo surgery and postoperative radiation therapy. The difference in treatment outcomes may be attributed to worse clinical features, such as extensive disease and involvement of cranial nerves, rather than treatment modality. As patients with ACC tend to survive for many years, the balance between risk and benefit needs to be thoroughly considered before determining the initial treatment.

**Abstract:**

This study aimed to present the treatment patterns and outcomes for adenoid cystic carcinoma (ACC) arising in the nasal cavity and paranasal sinus. Sixty-one sinonasal ACC patients were retrospectively reviewed: 31 (50.8%) underwent surgery followed by postoperative radiation therapy (S+PORT), and 30 (49.2%) received definitive radiation therapy (D(C)RT). T4 disease was significantly more frequent in the D(C)RT group (25.8% vs. 80.0%, *p* < 0.001), where all T4b disease patients underwent D(C)RT. The 5-year local failure-free survival (LFFS), distant metastasis-free survival (DMFS), progression-free survival (PFS), and overall survival were 61.8% versus 37.8% (*p* = 0.003), 64.8% versus 38.1% (*p* = 0.036), 52.6% versus 19.3% (*p* = 0.010), and 93.2% versus 73.4% (*p* = 0.001) in the S+PORT and D(C)RT groups, respectively. The absolute differences in 5-year rates of LFFS, DMFS, and PFS between the two groups were smaller in the T3–4 subgroup. The univariate analysis showed that T4b disease, neurologic symptoms, longest diameter of tumor, radiological evidence of nerve involvement, and undergoing D(C)RT were associated with worse clinical outcomes, but the significance disappeared in the multivariate analysis, except for in the case of radiological evidence of nerve involvement. In conclusion, most patients with extensive disease underwent upfront D(C)RT and generally exhibited inferior clinical outcomes when compared to those with less extensive disease and who underwent S+PORT.

## 1. Introduction

Adenoid cystic carcinoma (ACC) is a rare histologic type that originates from the glandular tissues throughout the body [1]. ACC can involve various anatomical sites [2] and comprises about 1% of all head and neck malignancies [3]. Although ACC ranks among the most common malignant tumors affecting the major and minor salivary glands [4], its occurrence in the sinonasal region is significantly less frequent, constituting approximately 6–7% of cases according to database studies [5,6]. Despite its relatively indolent growth characteristic, managing ACC in the head and neck region poses some challenges due to its locally persistent and recurrent clinical pattern, as well as the frequent and delayed onset of distant metastases, particularly in the lungs [4].

Surgery stands as the standard treatment modality for patients with head and neck ACC [4]. Postoperative radiation therapy (PORT) is commonly recommended to reduce the risk of local and regional recurrence, and several previous reports have demonstrated improved local control by adding PORT [7,8]. However, the tendency of ACC to infiltrate through the nerve pathways, and the proximity of the nasal cavity and paranasal sinuses to the skull base, present unique challenges. Sinonasal malignancies are frequently diagnosed at locally advanced stages [6]. While neoadjuvant chemotherapy followed by radical surgery is a viable option for patients with squamous cell carcinoma histology [9], ACC usually displays limited responsiveness to chemotherapy, leaving no established standard chemotherapy regimen as of yet [10]. In cases of unresectable ACC in the head and neck, definitive radiation therapy with or without concurrent chemotherapy (D(C)RT) has been a common approach. Comparative studies have indicated that upfront D(C)RT yields less favorable treatment outcomes when compared to upfront radical surgery [6,11,12]; however, the interpretation of these results requires caution due to variations in baseline patient characteristics. This study aimed to present the treatment patterns for ACC arising in the nasal cavity and paranasal sinus and to compare the treatment outcomes following D(C)RT and surgery followed by PORT (S+PORT).

## 2. Materials and Methods

### 2.1. Patient Population

We retrospectively reviewed the medical records of the patients who were diagnosed as having pathologically confirmed ACC of the nasal cavity and paranasal sinuses and were treated between 1995 and 2021 at the authors’ institute. Patients who had metastatic disease at the time of diagnosis or were treated for recurrent disease were excluded. Patients who underwent radiation therapy (RT) at other institutions or did not receive RT were also excluded. The eligible patients were categorized into two groups based on upfront treatment modalities: those who underwent upfront radical surgery followed by PORT (the S+PORT group), and those who received upfront D(C)RT without surgery (the D(C)RT group), respectively.

### 2.2. Treatment

The established treatment approach for the resectable sinonasal ACC patients at our institute has been radical surgery, and PORT is usually considered when the post-surgical pathology reports reveal risk factor(s), which typically includes positive or close (<0.5 cm) resection margins, pT3-4 disease, or perineural invasion, respectively. Most of the D(C)RT group patients had an unresectable disease extent, as evaluated by the head and neck surgeons. Other reasons for opting against radical surgery included the anticipated significant functional impairment and disfigurement following upfront surgery, medical comorbidities, and patients’ preferences, respectively.

A few RT techniques were employed along with the technical advancements over the past 26 years’ study period, which included 3-dimensional conformal RT (3D-CRT), intensity-modulated RT (IMRT), and proton beam therapy (PBT). All patients underwent CT simulation while under the thermoplastic mask to delineate the target volumes and organs at risk (OARs). The gross tumor volume (GTV) was encompassed by the primary disease identified through the radiographic images and physical examination. The GTV delineation was omitted in the case of the PORT setting when there was no evidence of residual disease. The clinical target volume (CTV) included the area immediately adjacent to the GTV or the resection bed, taking into account the potential for microscopic invasion. Elective nodal irradiation of the neck was not routinely conducted. The planning target volume (PTV) was generated by expanding the CTV by 3 mm in all directions. The prescribed dose schedules were determined on an individual patient basis, mainly considering the purpose of RT (definitive vs. postoperative) and the estimated local recurrence risk. A simultaneous integrated boost technique was applied to differentiate the dose to the GTV and CTV for the patients undergoing IMRT and/or PBT. Both conventional fractionation (2 Gy per fraction) and moderate hypofractionation (up to 3 Gy per fraction) were employed depending on the RT purpose and techniques. RT was administered once daily, five fractions per week in all patients. In the D(C)RT group, selected patients underwent concurrent chemotherapy with intravenous cisplatin (100 mg/m^2^), administered starting on the first day of RT at three-week intervals. All patients were regularly evaluated in the outpatient clinic at intervals of 3–4 months during the first 2 years, 6 months until the fifth year, and once a year thereafter, respectively. These evaluations encompassed history-taking, physical examination, laboratory tests, and radiological examinations, as needed.

### 2.3. Endpoints and Statistical Analysis

The clinical outcomes subjected to the current analysis included local failure-free survival (LFFS), distant metastasis-free survival (DMFS), progression-free survival (PFS), and overall survival (OS), respectively. For LFFS, the event was defined as the manifestation of local progression at the primary site within the RT target volume, which included the emergence or progression of perineural spread lesions in the adjacent region, or deaths of patients. The event for DMFS was defined as either the occurrence of distant metastasis or the emergence of new intracranial lesions not contiguous with the previous RT target volume, or deaths of patients. The PFS events were defined as any form of disease progression or metastasis, or deaths of patients. The OS events were defined as the deaths of patients from any cause. All durations were measured from the initiation of the upfront treatment to the following events: the date of surgery in the S+PORT group, and the date of RT initiation in the D(C)RT group, respectively. The Kaplan–Meier method was utilized for calculating the survival rates, and the log-rank test was used for comparing these outcomes. The uni- and multivariate analyses were conducted to assess the potential variables that could impact the clinical outcomes, utilizing the Cox proportional hazards model. Variables with a *p*-value of <0.1 in the univariate analyses were included in the subsequent multivariate analyses. The treatment-related adverse events were defined as symptoms that occurred or worsened during or after treatment. Meanwhile, the symptoms that occurred or worsened after local progression and were linked with progression were not included in the adverse events. The events graded ≥3 in accordance with the Common Terminology Criteria for Adverse Events (CTCAE) version 5.0 were recorded. The crude rates of occurrence of grade ≥3 events were calculated for both groups. The statistical significances of differences were assessed using the chi-square test for the categorical variables and the Student’s *t*-test for the continuous variables, respectively. In all statistical tests, a *p*-value of <0.05 was considered indicative of statistical significance. All statistical analyses were performed using the R software (version 4.2.1; The R Foundation for Statistical Computing, Vienna, Austria).

## 3. Results

### 3.1. Patients’ Characteristics and Treatment Specifics

A total of 61 eligible patients were included in the current study: 31 (50.8%) were allocated to the S+PORT group, and 30 (49.2%) were allocated to the D(C)RT group, respectively. The patients’ characteristics and treatment specifics are summarized in Table 1, and the median follow-up period was 5.3 years (range: 0.8–27.4 years). The reasons for choosing D(C)RT, instead of surgery, were unresectable disease extent at the time of diagnosis in 22 patients (73.3%), patient’s preference for D(C)RT in five (16.7%), severe anticipated functional impairment following surgical resection in two (6.7%), and poor performance status and significant medical comorbidities that precluded radical surgery in one (3.3%), respectively.

A substantial disparity was identified in the disease extent, according to the American Joint Committee on Cancer (AJCC) manual 8th edition, between the following groups: T4 disease constituted 80% of the D(C)RT group patients, predominantly featuring T4b disease, while it constituted 25.8% of the S+PORT group patients, where none had T4b disease (*p* < 0.001), respectively. No patient in this study presented with regional lymph node metastasis. In terms of tumor size, measured through the diagnostic imaging studies, the D(C)RT group patients had larger tumors (median longest diameter: 3.7 cm vs. 4.2 cm, *p* = 0.033). A higher incidence of radiological evidence indicating nerve involvement, defined as infiltration into known nerve pathways such as the skull base foramina or perineural invasion, was apparent in the D(C)RT group (80.0% vs. 32.3%, *p* < 0.001). Overall, the D(C)RT group displayed a greater prevalence of a more advanced tumor extent when compared to the S+PORT group. A significant difference was observed in the total radiation dose between the groups (the median equivalent dose in 2 Gy fractions: 60.4 Gy vs. 67.7 Gy, *p* < 0.001), which reflected the different aims of RT in the two groups.

### 3.2. Clinical Outcomes and Patterns of Recurrences

The clinical outcomes of all patients based on the treatment groups are illustrated in Figure 1. The numbers of LFFS, DMFS, PFS, and OS events observed through this study were 36, 36, 45, and 21, respectively. The patterns of recurrence according to the T stage and treatment group are summarized in Table 2, and the patients with T4b disease exhibited the highest rates of any progression (83.3%) and local progression (72.2%). Among the 33 patients who experienced distant metastasis, the lung was most frequently involved (26, 78.8%). Six out of the seven patients who did not develop lung metastasis initially had T4 disease, where four developed intracranial metastasis and three developed bone metastasis, respectively.

Among all patients, the D(C)RT group patients, in comparison to their S+PORT counterparts, exhibited significantly lower 5-year rates of LFFS, DMFS, PFS, and OS, respectively (Table 3). In the T1–2 patients, comparisons of clinical outcomes between the treatment groups were not feasible because only two patients underwent D(C)RT. The clinical outcomes of the patients with T3–4 tumors in the two groups are presented in Figure 2. Among the 45 T3–4 patients, between the treatment groups, the difference in LFFS, DMFS, and PFS turned out to be insignificant; however, there remained a significant difference in OS (Table 3). The absolute difference in 5-year rates of clinical outcomes between the two groups was smaller in the T3–4 subgroup (24.0–33.3% versus 6.6–18.2%), except in terms of OS. Among the patients who experienced local progression or distant metastasis, the median durations until local progression and distant metastasis were 3.2 years (range: 0.7–14.3 years) and 3.8 years (range: 0.3–14.4 years), respectively. Among the patients who died following local progression, the median duration from local progression to death was 3.3 years (range: 0.2–10.0 years).

Among the patients who underwent upfront D(C)RT, the addition of concurrent chemotherapy to RT did not affect any clinical outcomes (LFFS: hazard ratio [HR] 0.982, 95% CI 0.325–2.967, *p* = 0.974; DMFS: HR 1.833, 95% CI 0.640–5.256, *p* = 0.259; PFS: HR 1.857, 95% CI 0.709–4.865, *p* = 0.208; OS: HR 1.356, 95% CI 0.366–5.031, *p* = 0.649). Among the patients who underwent S+PORT, the margin status again was not associated with any clinical outcomes (LFFS: HR 3.391, 95% CI 0.929–12.37, *p* = 0.065; DMFS: HR 1.295, 95% CI 0.439–3.819, *p* = 0.640; PFS: HR 1.697, 95% CI 0.607–4.744, *p* = 0.313; OS: HR 1.671, 95% CI 0.300–9.299, *p* = 0.558).

### 3.3. Univariate and Multivariate Analyses

The results of the uni- and multivariate analyses conducted for the clinical outcomes are summarized in Table 4. In the univariate analysis, four factors were unfavorably associated with at least one outcome: T4b disease was with LFFS (*p* < 0.001), PFS (*p* = 0.013), and OS (*p* < 0.001); neurologic symptoms as chief complaints with LFFS (*p* = 0.036) and OS (*p* = 0.033); longest diameter of tumor in radiological examination with OS (*p* = 0.018); radiological evidence of nerve involvement with LFFS (*p* = 0.010), DMFS (*p* = 0.009), PFS (*p* < 0.001), and OS (*p* = 0.046); and D(C)RT group with LFFS (*p* = 0.003), DMFS (*p* = 0.038), PFS (*p* = 0.012), and OS (*p* = 0.001), respectively. In the multivariate analyses, however, only radiological evidence of nerve involvement was associated with poor DMFS (*p* = 0.042) and PFS (*p* = 0.004).

### 3.4. Adverse Events

No grade 4–5 events were observed during the current follow-up, while grade 3 events were observed in six patients (19.4%) among the S+PORT group and seven (23.3%) among the D(C)RT group, respectively (Table 5). There was no significant difference in the crude rates of grade 3 adverse events between the groups (*p* = 0.947).

## 4. Discussion

This study presents the clinical outcomes of patients with ACC arising in the nasal cavity and paranasal sinus who were diagnosed and managed within a single institution over the time span of 26 years. Most patients with resectable disease and favorable conditions underwent upfront radical surgery followed by PORT, whereas those with unresectable disease and unfavorable conditions received upfront D(C)RT. The D(C)RT group patients exhibited generally worse clinical outcomes when compared to their S+PORT counterparts, which contrasts with the findings of a study performed by the authors which included only patients with sinonasal squamous cell carcinoma [13]. The differences in the baseline patient characteristics, particularly the disease extent, however, should not be disregarded. In the D(C)RT group, 60% of the patients displayed T4b disease, according to the AJCC 8th staging, implying that the critical structures of the skull base, well-known cranial nerve invasion pathways, were already involved. Given the tendency of ACC to spread perineurally and the prognostic relevance of this [14,15], the difficult challenges in controlling this situation are well-anticipated. The disparities in the clinical outcomes were mitigated by the multivariate analyses, which included both the disease extent and the treatment groups, and by focusing only on the patients with T3–4 disease. Based on these findings, it can be speculated that the inferior clinical outcomes observed in the D(C)RT group were mostly influenced by the initial disease extent, which signified the considerable challenges in treating the T4b ACC patients with upfront RT.

In this study, all T4b patients underwent D(C)RT, and the vast majority (83.3%) encountered some form of disease progression subsequently, mainly local progression (72.2%). Although promising outcomes were reported for selected T4b patients who underwent extensive surgical resection [16], the vast majority of T4b patients tend to undergo D(C)RT in the real world, due to the unresectable extent of the disease and concerns about functional deficits and cosmetic demerits following upfront surgical resection. Even among patients who are deemed resectable, achieving adequate and satisfactory surgical margins seems elusive, frequently necessitating PORT following surgery. Consequently, efforts have been directed toward enhancing the efficacy of RT in treating T4b patients with skull-base involvement. Particle beam therapy could arise as one of the promising options in this regard. The distinct physical properties of proton or carbon ion beams may confer dosimetric advantages in treating skull base tumors, notably in terms of better sparing the OARs [17,18]. The key to successful RT basically lies in how to maintain target volume coverage while optimally sparing the critical OARs, which can be more favorably achieved by employing particle beam therapy. Previous studies exhibited promising local control with particle beam therapy for sinonasal and nasopharyngeal ACC [19,20]. However, a French study that investigated T4 sinonasal ACC with incomplete surgical resection (R1 or R2) or non-operated cases reported a 3-year local control rate of 60%, underscoring the complexities tied to achieving local control in locally advanced sinonasal ACC [21]. While particle beam therapy may confer benefits for some selected patients when compared to photon therapy, treating sinonasal ACC patients with extensive skull-base involvement with particle beam therapy still presents great challenges. Even with particle beam therapy, sparing the critical structures located close to the target volumes frequently remains a demanding task and constrains the potential for escalated-dose delivery that is desired for better local disease control. In this study, approximately 30% of patients underwent PBT alone or in combination with photon IMRT. Currently, our institution optionally offers PBT for sinonasal ACC patients, on the condition that dosimetric advantages are expected over photon IMRT alone.

Another approach for enhancing RT efficacy is concurrent chemotherapy. While the chemotherapy efficacy remains limited in ACC [10], cisplatin concurrently delivered with RT has demonstrated synergy with radiation [22]. Its role in the concurrent chemotherapy setting is closer to that of a radiosensitizer, rather than that of a cytotoxic agent. Promising outcomes following concurrent chemoradiation in this setting, however, were often underscored by virtue of the small sample sizes [23,24]. Moreover, in treating highly advanced (T4b) ACC patients with concurrent chemoradiation, the clinical outcomes have generally not been very favorable [25,26]. As yet, comparative studies with sufficiently large patient cohorts are not available to thoroughly endorse the role of concurrent chemotherapy in managing ACC patients, which still remains a debatable subject. In this study, approximately 20% of the D(C)RT group patients underwent concurrent chemoradiation, and the subgroup comparison showed no difference in the clinical outcomes by adding chemotherapy to RT, which might presumably be due to not enough patient numbers. Concurrent chemoradiation with tri-weekly cisplatin, however, has been actively employed at our institution, and a more comprehensive report will be available when a sufficiently large patient cohort is accrued in the future.

The multivariate analysis conducted in this study did not reveal any statistically significant associations between the variables and the outcomes, except for the radiological evidence of nerve involvement. This may be partly attributed to the relatively small patient cohort size. Nevertheless, the univariate analysis showed several notable factors. The impact of cranial nerve involvement [14,15] and skull base invasion [12,27] as worse prognostic factors for ACC has been well-established and has been reaffirmed through our study. In a previous analysis for lacrimal gland ACC, conducted by the authors, the patients with gross residual tumor burden following surgical resection demonstrated worse outcomes, which signified the importance of tumor burden at the time of D(C)RT [28]. In the current study, however, the initial tumor size was associated with worse OS only, whereas T4b disease was associated with worse LFFS, PFS, and OS, respectively. Additionally, radiological evidence of nerve involvement demonstrated the associations with all four clinical outcomes. The infiltration of the tumor into the critical structures and major nerves may hold greater prognostic significance than the initial tumor burden for sinonasal ACC, possibly due to its proximity to the skull base structures.

The distinctive clinical behavior of ACC is characterized by the relatively slow and consistent disease progression, and this study likewise highlighted these traits. Through the current study observation, disease progression events frequently manifested several years following treatment, with some instances spanning over a decade. Even following local progression, it often took several more years before the patients succumbed to death. Similar clinical patterns were observed in other studies with long-term follow-ups [29,30]. The clinicians should be aware of this distinct clinical behavior and ensure that the patients are monitored for a sufficiently longer time for follow-up evaluation.

There are a few limitations of this study. First, the imbalanced baseline characteristics between the treatment groups seem to have hindered the drawing of concrete conclusions about the efficacy of specific treatment modalities. Second, due to the retrospective nature, the events for the clinical outcomes and toxicities could have been underreported. Third, the relatively small cohort size might have restricted the statistical power. Despite these limitations, this study could contribute a few valuable clinical insights into the management of sinonasal ACC patients, which is particularly significant given the rarity of this disease.

## 5. Conclusions

For sinonasal ACC, most patients with extensive disease involvement underwent upfront D(C)RT and demonstrated inferior clinical outcomes when compared to those with less extensive disease and who underwent upfront S+PORT. The initial tumor burden (T4b disease) seems to be a notable challenge when applying upfront D(C)RT. The balance between risk and benefit needs to be thoroughly considered before determining the initial treatment modality, as many patients tend to survive for many years regardless of the initial tumor burden. The development of novel treatment approaches that could enhance RT effectiveness and lead to more favorable outcomes is highly desirable.

## Figures and Tables

**Figure 1 cancers-16-01235-f001:**
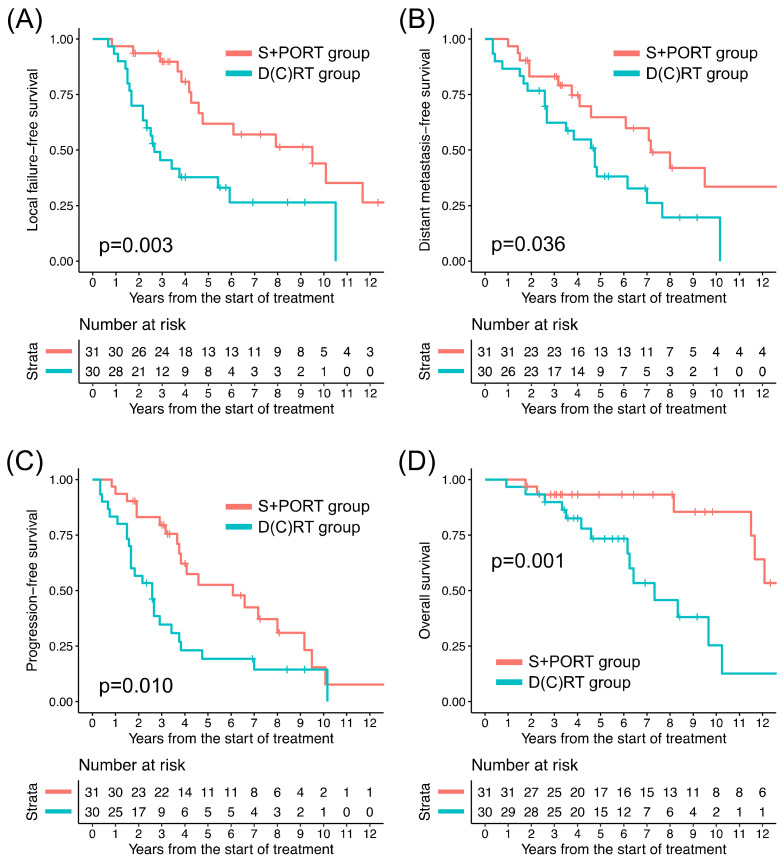
Kaplan–Meier curves of (**A**) local failure-free survival, (**B**) distant metastasis-free survival, (**C**) progression-free survival, and (**D**) overall survival of two treatment groups for all patients. D(C)RT: definitive radiation therapy ± concurrent chemotherapy, S+PORT: surgery followed by postoperative radiation therapy.

**Figure 2 cancers-16-01235-f002:**
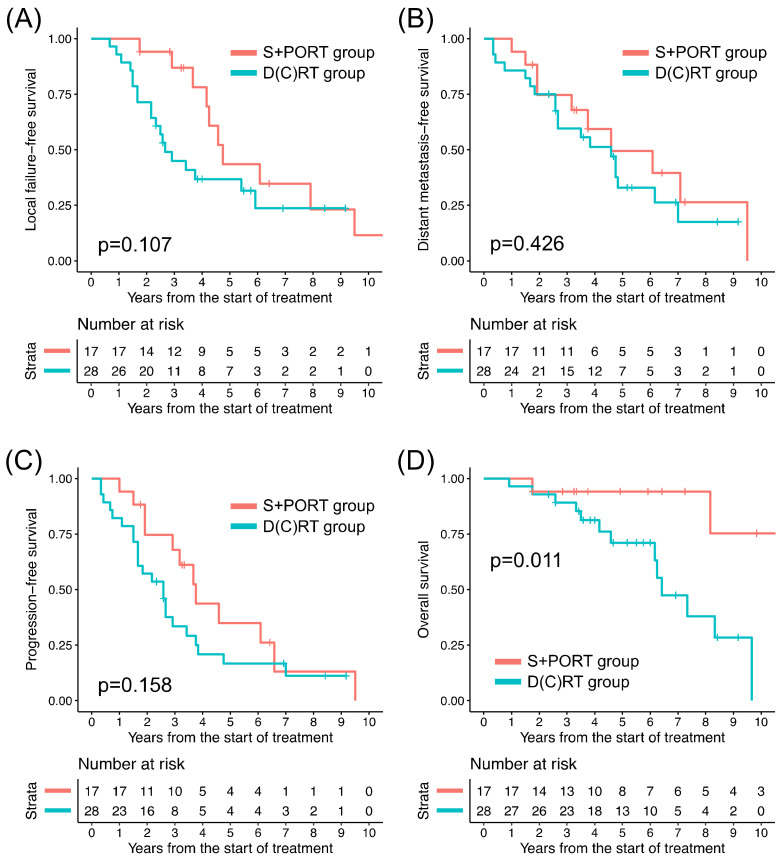
Kaplan–Meier curves of (**A**) local failure-free survival, (**B**) distant metastasis-free survival, (**C**) progression-free survival, and (**D**) overall survival of two treatment groups for T3–4 disease. D(C)RT: definitive radiation therapy ± concurrent chemotherapy, S+PORT: surgery followed by postoperative radiation therapy.

**Table 1 cancers-16-01235-t001:** Patients’ characteristics and treatment specifics.

Characteristics	S+PORT Group (N = 31)	D(C)RT Group (N = 30)	*p*-Value
Age at diagnosis (years, median)	48 (range: 28–67)	54 (range: 20–82)	0.168
Sex			1.000
Male	15 (48.4%)	15 (50.0%)	
Female	16 (51.6%)	15 (50.0%)	
Time of treatment initiation			0.554
1995–1999	1 (3.2%)	3 (10.0%)	
2000–2004	3 (9.7%)	2 (6.7%)	
2005–2009	5 (16.1%)	2 (6.7%)	
2010–2014	5 (16.1%)	7 (23.3%)	
2015–2019	11 (35.5%)	13 (43.3%)	
2020–2021	6 (19.4%)	3 (10.0%)	
Primary site			0.074
Nasal cavity	14 (45.2%)	10 (33.3%)	
Maxillary sinus	17 (54.8%)	14 (46.7%)	
Ethmoid sinus	0 (0.0%)	2 (6.7%)	
Sphenoid sinus	0 (0.0%)	4 (13.3%)	
T stage (AJCC 8th)			**<0.001**
T1	3 (9.7%)	0 (0.0%)	
T2	11 (35.5%)	2 (6.7%)	
T3	9 (29.0%)	4 (13.3%)	
T4a	8 (25.8%)	6 (20.0%)	
T4b	0 (0.0%)	18 (60.0%)	
Neurologic symptom as chief complaints	1 (3.2%)	13 (43.3%)	**0.001**
PET/CT staging	26 (83.9%)	25 (83.3%)	1.000
Longest diameter on radiologic exams (cm, median)	3.7 (range: 2.0–5.5) *	4.2 (range: 1.6–6.4) *	**0.033**
Longest diameter on pathologic exam (cm, median)	4.0 (range; 2.0–8.0) ^†^	-	-
Solid portion			0.741 ^‡^
Yes	4 (12.9%)	3 (10.0%)	
No	21 (67.7%)	8 (26.7%)	
Unknown	6 (19.4%)	19 (63.3%)	
Radiological evidence of nerve involvement ^§^	10 (32.3%)	24 (80.0%)	**<0.001**
Perineural invasion in pathologic exam			-
Yes	13 (41.9%)	-	
No	8 (25.8%)	-	
Unknown	10 (32.3%)	-	
Margin status			-
Positive (R1)	20 (64.5%)	-	
Close (<0.5 cm)	11 (35.5%)	-	
Neck dissection	8 (25.8%)	-	-
Concurrent chemotherapy	0 (0.0%)	7 (23.3%)	**0.014**
Radiation therapy modality			0.489
3D-CRT	7 (22.6%)	6 (20.0%)	
IMRT	12 (38.7%)	16 (53.3%)	
Proton therapy (alone or combined with IMRT)	12 (38.7%)	8 (26.7%)	
Radiation dose, EQD2 (Gy, median) ^¶^	60.4 (range: 50.0–70.0)	67.7 (range: 57.3–71.6)	**<0.001**
Radiation therapy duration (days, median)	41 (range: 34–47)	41 (range: 29–53)	0.580

* One patient in each group did not have available radiologic exams for evaluation. ^†^ Pathologic reports of four patients did not include the size of the tumor. ^‡^
*p*-values were calculated excluding patients with an unknown variable. ^§^ The definition of radiological evidence of nerve involvement was evidence of infiltration into known nerve pathways such as foramina in the skull base or perineural invasion in the radiological exams. ^¶^ EQD2 was calculated with an alpha-beta ratio of 10. Treatment duration was not considered. *p*-values that were statistically significant (<0.05) were marked in bold. Abbreviations: 3D-CRT, three-dimensional conformal radiation therapy; AJCC, American Joint Committee on Cancer; D(C)RT, definitive radiation therapy ± concurrent chemotherapy; EQD2, equivalent dose in 2 Gy fractions; IMRT, intensity-modulated radiation therapy; PET/CT, positron emission tomography/computed tomography; S+PORT, surgery followed by postoperative radiation therapy.

**Table 2 cancers-16-01235-t002:** Patterns of recurrence according to the T stage and treatment groups.

Patterns of Recurrence	T1	T2		T3		T4a		T4b
	S+PORT Group (N = 3)	S+PORT Group (N = 11)	D(C)RT Group (N = 2)	S+PORT Group (N = 9)	D(C)RT Group (N = 4)	S+PORT Group (N = 8)	D(C)RT Group (N = 6)	D(C)RT Group (N = 18)
Any progression	0 (0.0%)	8 (72.7%)	2 (100.0%)	7 (77.8%)	1 (25.0%)	5 (62.5%)	6 (100.0%)	15 (83.3%)
Local progression	0 (0.0%)	4 (36.4%)	2 (100.0%)	6 (66.7%)	0 (0.0%)	3 (37.5%)	4 (66.7%)	13 (72.2%)
Regional recurrence	0 (0.0%)	1 (9.1%)	0 (0.0%)	0 (0.0%)	0 (0.0%)	1 (12.5%)	1 (16.7%)	2 (11.1%)
Distant metastasis	0 (0.0%)	5 (45.5%)	2 (100.0%)	5 (55.6%)	1 (25.0%)	5 (62.5%)	5 (83.3%)	10 (55.6%)

Abbreviations: D(C)RT: definitive radiation therapy ± concurrent chemotherapy, S+PORT: surgery followed by postoperative radiation therapy.

**Table 3 cancers-16-01235-t003:** Comparison of clinical outcomes in all patients and T3–4 subgroup.

	5-Year Rate of LFFS (95% CI)		5-Year Rate of DMFS (95% CI)		5-Year Rate of PFS (95% CI)		5-Year Rate of OS (95% CI)	
	S+PORT	D(C)RT	S+PORT	D(C)RT	S+PORT	D(C)RT	S+PORT	D(C)RT
All patients	61.8%	37.8%	64.8%	38.1%	52.6%	19.3%	93.2%	73.4%
	(44.6%–85.6%)	(23.5%–60.8%)	(48.2%–87.1%)	(23.4%–62.3%)	(35.9%–77.2%)	(8.9%–41.8%)	(84.5%–100%)	(58.0%–92.8%)
	*p* = 0.003		*p* = 0.036		*p* = 0.010		*p* = 0.001	
T3–4 subgroup	43.4%	36.8%	49.5%	32.9%	34.9%	16.7%	94.1%	71.1%
	(22.7%–83.2%)	(22.2%–60.9%)	(28.2%–86.8%)	(18.5%–58.7%)	(16.4%–74.4%)	(6.9%–40.3%)	(83.6%–100%)	(54.7%–92.2%)
	*p* = 0.107		*p* = 0.426		*p* = 0.158		*p* = 0.011	

Abbreviations: CI, confidence interval; D(C)RT, definitive radiation therapy ± concurrent chemotherapy; DMFS, distant metastasis-free survival; LFFS, local failure-free survival; PFS, progression-free survival; OS, overall survival; S+PORT, surgery followed by postoperative radiation therapy.

**Table 4 cancers-16-01235-t004:** Univariate and multivariate analyses for the clinical outcomes.

Characteristics (Comparison vs. Reference)	Local Failure-Free Survival						Distant Metastasis-Free Survival						Progression-Free Survival						Overall Survival					
	Univariate			Multivariate			Univariate			Multivariate			Univariate			Multivariate			Univariate			Multivariate		
	HR	95% CI	*p*-Value	HR	95% CI	*p*-Value	HR	95% CI	*p*-Value	HR	95% CI	*p*-Value	HR	95% CI	*p*-Value	HR	95% CI	*p*-Value	HR	95% CI	*p*-Value	HR	95% CI	*p*-Value
Age at diagnosis (per a year)	1.027	0.998–1.056	0.070	1.022	0.993–1.052	0.137	1.021	0.992–1.051	0.150	-	-	-	1.021	0.996–1.047	0.106	-	-	-	1.041	0.999–1.084	0.057	1.038	0.995–1.083	0.084
Sex (female vs. male)	0.992	0.513–1.918	0.980	-	-	-	1.086	0.564–2.094	0.804	-	-	-	1.511	0.820–2.782	0.186	-	-	-	0.779	0.328–1.853	0.573	-	-	-
Primary site (maxillary sinus vs. others)	0.664	0.341–1.295	0.229	-	-	-	1.688	0.859–3.318	0.129	-	-	-	0.939	0.513–1.719	0.838	-	-	-	0.987	0.417–2.336	0.976	-	-	-
T4b disease (yes vs. no)	3.586	1.751–7.347	**<0.001**	2.024	0.596–6.870	0.258	1.988	0.968–4.081	0.061	0.871	0.334–2.270	0.777	2.250	1.189–4.255	**0.013**	0.567	0.217–1.487	0.249	7.702	2.609–22.74	**<0.001**	4.829	0.598–33.44	0.111
Neurologic symptom as chief complaints (yes vs. no)	2.189	1.053–4.548	**0.036**	0.807	0.306–2.127	0.664	1.424	0.678–2.992	0.350	-	-	-	1.454	0.743–2.844	0.274	-	-	-	2.684	1.086–6.635	**0.033**	0.605	0.148–2.464	0.483
PET/CT staging (yes vs. no)	1.064	0.472–2.401	0.881	-	-	-	1.331	0.585–3.028	0.495	-	-	-	2.399	0.913–6.301	0.076	1.715	0.620–4.745	0.299	0.780	0.300–2.030	0.611	-	-	-
Longest diameter on radiologic exam (per 1 cm) *	1.265	0.933–1.717	0.130	-	-	-	1.153	0.879–1.512	0.305	-	-	-	1.192	0.923–1.539	0.178	-	-	-	1.588	1.081–2.333	**0.018**	1.281	0.897–1.867	0.197
Radiological evidence of nerve involvement (yes vs. no)	2.634	1.255–5.526	**0.010**	1.601	0.641–4.001	0.314	2.652	1.275–5.517	**0.009**	2.369	1.032–5.437	**0.042**	3.741	1.838–7.615	**<0.001**	3.478	1.501–8.063	**0.004**	2.612	1.016–6.717	**0.046**	0.822	0.202–3.351	0.785
Radiation therapy modality																								
IMRT vs. 3D-CRT	1.108	0.484–2.538	0.808	-	-	-	1.121	0.523–2.403	0.769	-	-	-	1.602	0.747–3.435	0.226	-	-	-	0.696	0.273–1.776	0.449	-	-	-
Proton therapy (alone or combined with IMRT) vs. 3D-CRT	1.331	0.494–3.585	0.572	-	-	-	1.228	0.440–3.425	0.695	-	-	-	1.402	0.550–3.576	0.480	-	-	-	0.616	0.116–3.287	0.571	-	-	-
Treatment group (D(C)RT vs. S+PORT group)	2.878	1.416–5.846	**0.003**	1.717	0.644–4.579	0.280	2.077	1.042–4.139	**0.038**	1.672	0.696–4.013	0.250	2.172	1.189–3.967	**0.012**	2.143	0.929–4.939	0.074	5.093	1.872–13.86	**0.001**	3.390	0.845–13.59	0.085

* Two patients did not have available radiologic exams for evaluation. *p*-values that were statistically significant (<0.05) were marked in bold. Abbreviations: 3D-CRT, three-dimensional conformal radiation therapy; CI, confidence interval; D(C)RT, definitive radiation therapy ± concurrent chemotherapy; HR, hazard ratio; IMRT, intensity-modulated radiation therapy; PET/CT, positron emission tomography/computed tomography; S+PORT, surgery followed by postoperative radiation therapy.

**Table 5 cancers-16-01235-t005:** The details of the grade 3 adverse events.

Patient Number	Treatment Group	Sex	Age	Primary Site	T Stage	CTCAE Terminology	Months from Treatment Initiation	Course
#1	S+PORT	F	51	Maxillary sinus	T4a	Sinusitis	26	Surgical debridement was required.
#2	S+PORT	F	66	Maxillary sinus	T2	Facial pain	34	Severe facial pain due to a titanium plate inserted during the previous radical surgery. Partial removal of the plate was indicated.
						Skin ulceration	54	Partial removal of the titanium plate and a skin graft was required.
#3	S+PORT	M	44	Maxillary sinus	T3	Osteonecrosis	35	Surgical debridement on the irradiated side of the maxilla was required.
#4	S+PORT	M	43	Nasal cavity	T2	Dermatitis radiation	1	Moist desquamation of the radiation field.
						Sinus disorder	6	Obstruction required surgical synechiolysis.
#5	S+PORT	F	48	Maxillary sinus	T2	Skin infection	3	Cellulitis of the treated site required hospitalization and intravenous antibiotics.
#6	S+PORT	F	59	Maxillary sinus	T3	Skin infection	4	Cellulitis of the treated site required hospitalization and intravenous antibiotics.
#7	D(C)RT	F	38	Sphenoid sinus	T4b	Extraocular muscle paresis	25	Diplopia during lateral gaze.
						Facial muscle weakness	28	Near complete facial palsy.
#8	D(C)RT	M	52	Maxillary sinus	T4b	Sinusitis	11	Surgical drainage was required.
						Hearing impaired	57	Hearing aid was indicated.
						Central nervous system necrosis	42	Progressive necrosis of the temporal lobe persisted after steroid administration. Bevacizumab was indicated.
						Extraocular muscle paresis	79	Diplopia during lateral gaze.
						Oral cavity fistula	83	Surgical closure was indicated.
#9	D(C)RT	F	60	Maxillary sinus	T3	Sinusitis	32	Surgery was recommended, but the patient refused.
#10	D(C)RT	F	64	Nasal cavity	T4a	Optic nerve disorder	37	Decreased visual acuity on the irradiated side.
#11	D(C)RT	F	69	Maxillary sinus	T4a	Sinusitis	3	Surgical debridement was required.
#12	D(C)RT	M	71	Maxillary sinus	T4b	Optic nerve disorder	23	Decreased visual acuity on the irradiated side.
#13	D(C)RT	F	64	Nasal cavity	T4b	Vestibular disorder	40	Severe dizziness necessitated an emergency department visit.

Abbreviations: CTCAE, Common Terminology Criteria for Adverse Events; D(C)RT, definitive radiation therapy ± concurrent chemotherapy; S+PORT, surgery followed by postoperative radiation therapy.

## Data Availability

The original contributions presented in the study are included in the article. Further inquiries can be directed to the corresponding author.

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
