# Peer review of "Clinical Outcomes in Adenoid Cystic Carcinoma of the Nasal Cavity and Paranasal Sinus: A Comparative Analysis of Treatment Modalities"

_cancers, 2024, doi:10.3390/cancers16061235_

Round 1

Reviewer 1 Report

Comments and Suggestions for Authors

This manuscript describes a single institution retrospective analysis of a relatively large population of patients with adenoid cystic ca in the paranasal sinuses. 

A good statistical analysis was performed and the outcome and correlations are well described. 

Some minor comments:

The surgery + Radiotherapy group they name PORT: I would change that to S+PORT group. The RT group also had Chemo in 20%. So change DRT in (C)RT.

Did resection margin status have an influence in the PORT group ? Add to the table.

Reviewer 2 Report

Comments and Suggestions for Authors

Abstract

·     95% confidence intervals should be added for all 5-year rates.

·         The contrasts of the overall treatment comparison with the T3-4 subset should be based on 5-year rates and/or hazard ratios, not p-values, especially given the very small sample sizes.

Materials and Methods

Endpoints and statistical analysis

·         Were regional progressions (as in table 2) included as an event for local control? If so, it would be more appropriate to call the endpoint “local-regional.” If not, where were they included?

·         Were deaths included as an event for progression-free rate? It appears based on figure 1 that they might have been. If so, this should be stated, and the endpoint should be called “progression-free survival.”

·         The Kaplan-Meier method is not appropriate when there are competing risks, in this case death without progression. The analysis should be redone calculating the LC, DMFR, and PFR rates by the cumulative incidence method, unless death is an event as mentioned above.

·         Selecting variables for multivariable analysis based on significance in univariate analysis is not appropriate. Selection based on an information criterion such AIC or BIC should be used.

·         The three subset analyses (T2-4a; T3-4; DRT only to evaluate chemotherapy) should be stated. It’s unclear why both T2-4a and T3-4 are necessary. A better way to illustrate the treatment effect in T stage subgroups would be to add an interaction term to the model and present treatment effect hazard ratios by T stage subgroup.

Results

Patients’ characteristics and treatment specifics 

·         Given the 26-year study period, consideration should be given to adding year treated to table 1, maybe in 5-year increments.

·         In the sentence starting “The reasons for choosing DRT…” the percentages sum to only 95.9.

Clinical outcomes and patterns of recurrences

·         The total number of events should be clearly stated for each endpoint.

·         95% confidence intervals should be added for all 5-year rates, medians, and hazard ratios.

·         The contrasts of the overall treatment comparison with the T3-4 subset should be based on 5-year rates and/or hazard ratios, not p-values, especially given the very small sample sizes.

Round 2

Reviewer 2 Report

Comments and Suggestions for Authors
